# PeerJ

# Formation of 1-octen-3-ol from *Aspergillus flavus* conidia is accelerated after disruption of cells independently of Ppo oxygenases, and is not a main cause of inhibition of germination

Kana Miyamoto[1], Tomoko Murakami[1], Pattana Kakumyan[1,3], Nancy P. Keller[2] and Kenji Matsui[1]

[1] Department of Biological Chemistry, Faculty of Agriculture and the Department of Applied Molecular Bioscience, Graduate School of Medicine, Yamaguchi University, Yamaguchi, Japan
[2] Departments of Bacteriology and Medical Microbiology/Immunology, University of Wisconsin-Madison, Madison, WI, USA
[3] Current affiliation: School of Science, Mae Fah Luang University, Chiang Rai, Thailand

## ABSTRACT

Eight-carbon (C8) volatiles, such as 1-octen-3-ol, are ubiquitous among fungi. They are the volatiles critical for aroma and flavor of fungi, and assumed to be signals controlling germination of several fungi. In this study, we found that intact *Aspergillus flavus* conidia scarcely synthesized C8 volatiles but repeated freeze-thaw treatment that made the cell membrane permeable promoted (*R*)-1-octen-3-ol formation. Loss or down regulation of any one of five fatty acid oxygenases (PpoA, PpoB, PpoC, PpoD or lipoxygenase) hypothesized contribute to 1-octen-3-ol formation had little impact on production of this volatile. This suggested that none of the oxygenases were directly involved in the formation of 1-octen-3-ol or that compensatory pathways exist in the fungus. Germination of the conidia was markedly inhibited at high density ($1.0 \times 10^9$ spores mL$^{-1}$). It has been postulated that 1-octen-3-ol is an autoinhibitor suppressing conidia germination at high density. 1-Octen-3-ol at concentration of no less than 10 mM was needed to suppress the germination while the concentration of 1-octen-3-ol in the suspension at $1.0 \times 10^9$ mL$^{-1}$ was under the detection limit ($<1$ μM). Thus, 1-octen-3-ol was not the principal component responsible for inhibition of germination. Instead, it was evident that the other heat-labile factor(s) suppressed conidial germination.

## INTRODUCTION

Eight-carbon (C8) volatiles, such as (*R*)-(-)-1-octen-3-ol, are found almost ubiquitously among fungi, and they are the characteristic of the fungal aroma (*Combet et al., 2006*). In fungi, they attract flies and mosquitoes, and in some fungi, they repel fungivory (*Combet et al., 2006*; *Brodhun & Feussner, 2011*). Interestingly, 1-octen-3-ol reduces

Corresponding author
Kenji Matsui,
matsui@yamaguchi-u.ac.jp

dopamine levels and causes dopamine neuron degeneration in *Drosophila melanogaster* (*Inamdar et al., 2013*).

C8 volatiles also cause responses in fungi. Conidiation of *Trichoderma* spp. was induced by C8 volatiles (*Nemcovic et al., 2008*). On the contrary, it was postulated that 1-octen-3-ol was a volatile autoinhibitor inhibiting unprofitable germination of *Penicillium paneum* conidia under harsh conditions, such as in highly crowded environments where competition for limited resources would be expected (*Chitarra et al., 2004*). The ability of C8 volatiles to regulate conidiation and germination of conidia was also reported with *Aspergillus nidulans* (*Herrero-Garcia et al., 2011*) and *Lecanicillium fungicola* (*Berendsen et al., 2013*). Because of these findings, it has been assumed that C8 volatiles, especially 1-octen-3-ol, perform signaling functions; however, no conclusive evidence supporting this hypothesis has been provided.

(*R*)-1-Octen-3-ol was derived from linoleic acid through dioxygenation to form linoleic acid (*S*)-10-hydroperoxide (HPO) and subsequent cleavage reaction to form (*R*)-1-octen-3-ol and 10-oxo-(*E*)-8-decenoic acid in mushrooms (*Agaricus bisporus*) (*Wurzenberger & Grosch, 1984*). However, the enzyme(s) involved in this pathway has not been identified yet. A group of fatty acid dioxygenases called Ppos [psi (precocious sexual inducer)-producing oxygenases] has been identified in fungi (*Brodhun & Feussner, 2011*). *Aspergillus nidulans* has three Ppos (PpoA, PpoB, and PpoC), and studies on the deletion mutants indicated that all the three Ppo enzymes are involved in Psi factor production (*Tsitsigiannis et al., 2005*). Recombinant *A. nidulans* PpoC showed an activity to form (*R*)-10-HPO of linoleic acid, and a portion of the HPO was further converted into C8 volatiles, such as 2-octen-1-ol, 2-octenal, 3-octanone and 1-octen-3-ol probably through non-enzymatic chemical fragmentation (*Brodhun et al., 2010*). Many fungi also have another type of fatty acid oxygenase, lipoxygenase (LOX) (*Brodhun & Feussner, 2011*). Ppos and LOXs are candidates for the enzyme involved in C8 volatile formation in fungi.

In this study, we investigated how C8 volatiles, especially, 1-octen-3-ol, are formed in conidia of *A. flavus. A. flavus* is an opportunistic pathogen of crops causing highly problematic infections because of aflatoxin contamination of seeds. *A. flavus* has four genes encoding Ppos, namely, *ppoA*, *ppoB*, *ppoC*, and *ppoD*, and one gene encoding LOX (*Brown et al., 2009*). We examined the formation of C8 volatiles in mutant lines of these fatty acid oxygenases. Furthermore, we quantified the amount of 1-octen-3-ol produced by conidia, and found that the endogenous amount was too low to be accountable for inhibition of germination of conidia.

## MATERIALS AND METHODS

### Strains and culture conditions

Wild type *A. flavus* strain, NRRL 3357, was used in this study. The *ppo* deletion strains, Δ*ppoA*, Δ*ppoC*, and Δ*ppoD*, and the LOX deletion strain, Δ*lox*, prepared by homologous recombination in the *pyrG⁻* strain NRRL 3357.5, and the RNAi mutant IRT4, depleted in expression of *lox* and all 4 *ppo* genes including *ppoB* were also used (*Brown et al., 2009*). *A. nidulans* (NRRL 1092) was obtained from Japan Collection of Microorganisms at Riken

Bioresource Center. All strains were grown at 29 °C on glucose minimal media (GMM) adjusted at pH 6.5 (*Shimizu & Keller, 2001*) unless otherwise indicated. Conidia suspensions were obtained from surface cultures incubated for 1 week by adding 10 mL of distilled water containing 0.02% (w/v) Tween 20 and by gentle mechanical removal with a sterile glass rod. The suspension was filtered through Miracloth (Calbiochem, La Jolla, CA).

## Volatile analysis

For sensitive detection of volatiles a solid phase microextraction (SPME)-GC/MS method was used, while for accurate quantification the volatiles were extracted with organic solvent for GC/MS analysis.

For SPME-GC/MS analysis, fiber coated with 50/30 µm DVB/Carboxen/PDMS Stable Flex (Supelco, Bellefonte, PA) was used. After incubating 300 µL of $1.0 \times 10^9$ mL$^{-1}$ conidial suspension in GMM for 9 h at 29 °C in a glass vial (20 mL), the vial was tightly capped. A portion of vials was kept at −20 °C for 1 h and thawed at 30 °C for 15 min. The fiber was exposed to the headspace of the vial at 22 °C for 30 min. Afterward, the fiber was transferred to an injection port of GC/MS (QP-5050; Shimadzu, Kyoto, Japan) equipped with a 0.25 mm ×30 m Stabiliwax column (Restek, Bellefonte, PA, USA), where compounds were desorbed at 200 °C for 1 min. The column temperature was 40 °C (5 min) to 200 °C (2 min) at 5 °C min$^{-1}$. The carrier gas (He) was at 1 mL min$^{-1}$. The mass detector was operated in the electron impact mode with ionization energy of 70 eV. To identify each compound, we used retention indices and MS profiles of corresponding authentic specimens.

For solvent extraction, the conidial suspension (300 µL) was mixed with 2 mL chloroform/methanol (1/2, v/v) containing 5 ng mL$^{-1}$ nonanyl acetate (as an internal standard). Thereafter, 0.4 mL chloroform and 0.75 mL of 1% (w/v) KCl were added into the mixture, vortexed, and centrifuged at 1000 rpm for 10 min. The organic layer was collected, and directly served to GC-MS analysis under the condition shown above. For resolution of enantiomers of 1-octen-3-ol, Alpha DEX 120 fused silica capillary column (0.25 mm × 30 m, Supelco) was used at a constant column temperature of 75 °C. Racemic 1-octen-3-ol was purchased from Alfa Aesar (Lancashire, UK). Its enantiomers were from Acros Organics (Geel, Belgium). Quantification was done with a calibration curve constructed with pure 1-octen-3-ol in GMM. The detection limit of 1-octen-3-ol was 1 µM with the signal to noise ratio more than 10. Heat inactivation was carried out by immersing the conidial suspension in a tightly sealed tube into boiling water for 10 min.

## Germination of conidia

The density of conidia was adjusted to be $1.0 \times 10^6$ to $1.0 \times 10^9$ mL$^{-1}$ in GMM containing 0.1% agar. The aqueous solution of 1-octen-3-ol was mixed with the conidial suspension when needed. The suspension (200 µL) was spread on a glass slide (26 × 76 mm), and incubated at 29 °C for 9 h in a humidified closed container. During incubation, the pH of medium did not change substantially. The conidia with the protrusion of a length longer than the diameter of conidia were counted as germinated. The length of hyphae was determined with an ImageJ software (http://rsbweb.nih.gov/ij/). Damaged

or dead cells were stained with Evans Blue (Wako Pure Chemicals, Osaka, Japan) (*Gaff & Okong'o-Ogola, 1971*). The dye [50 mg mL$^{-1}$ Evans Blue in 10% (w/v) Tween 20, 50 µL] was mixed with the equal volume of conidial suspension, incubated for 30 min, then, diluted four fold with distilled water for observation.

## Glucose content

*A. flavus* conidia were incubated in GMM for 9 h at 29 °C at $1.0 \times 10^9$ mL$^{-1}$, and the medium was cleared with a membrane filter (cellulose acetate 0.5 µm; Advantec, Tokyo, Japan). Glucose content in the medium was determined with a HPLC system equipped with GL-C610H column (10.7 mm i.d. × 300 mm; Hitachi High-Technologies Co., Tokyo, Japan). The mobile phase was water at 0.3 mL min$^{-1}$, and detection was done with a refractive index detector (L-2490; Hitachi High-Technologies).

## Evaluation of an autoinhibiting factor

In order to avoid the effect caused by deprivation of glucose in the medium after incubating conidia in GMM, the glucose content in the cleared medium was adjusted to 1% (w/v) by using the glucose content determined as above. A portion was heat-treated in boiling water for 10 min. Freshly prepared *A. flavus* conidia were suspended with fresh GMM, with the cleared, spent medium, or with the heat-treated cleared, spent medium at $1.0 \times 10^6$ mL$^{-1}$, and incubated for 9 h at 29 °C as described above to see the germination.

## Statistical analysis

Data obtained with at least in triplicate were analyzed using Excel-Toukei 2012 (SSRI, Tokyo, Japan). Germination of conidia was calculated using Kruskal-Wallis one-way ANOVA. Significant differences in their distribution at $P < 0.05$ are shown with different letters. The amounts of chemicals were calculated using Tukey multiple comparison test. Mean values with different letters are significant at $P < 0.05$.

# RESULTS

## Formation of carbon eight volatile compounds from *A. flavus* conidia

Volatiles were scarcely formed from intact *A. flavus* (NRRL 3357) conidia suspended in GMM at $1.0 \times 10^9$ mL$^{-1}$ (Fig. 1) and not detected in the headspace of conidiating *A. flavus* mycelium grown on GMM agar plate. On the contrary, volatiles were released when the suspension of conidia were treated with a freeze-thaw cycle. Among the C8 compounds, 1-octen-3-ol was most abundant, followed by 3-octanone, 2-octen-1-ol, and 1-octen-3-one (Fig. 1). Chiral-phase GC analyses indicated that *A. flavus* conidia formed (*R*)-(-)-1-octen-3-ol with an optical purity of >99% enantiomeric excess (Fig. 1, inset). When *A. flavus* conidia were treated with repeating freeze-thaw cycles, the amount of 1-octen-3-ol positively correlated with increased number of cycles (Fig. 2A). Evans Blue staining of the conidia indicated that the freeze-thaw treatment destroyed the plasma membrane and made them permeable to the pigment (Fig. 2C). The number of Evans Blue-positive cells roughly correlated with numbers of ungerminated cells (Fig. 2B) and

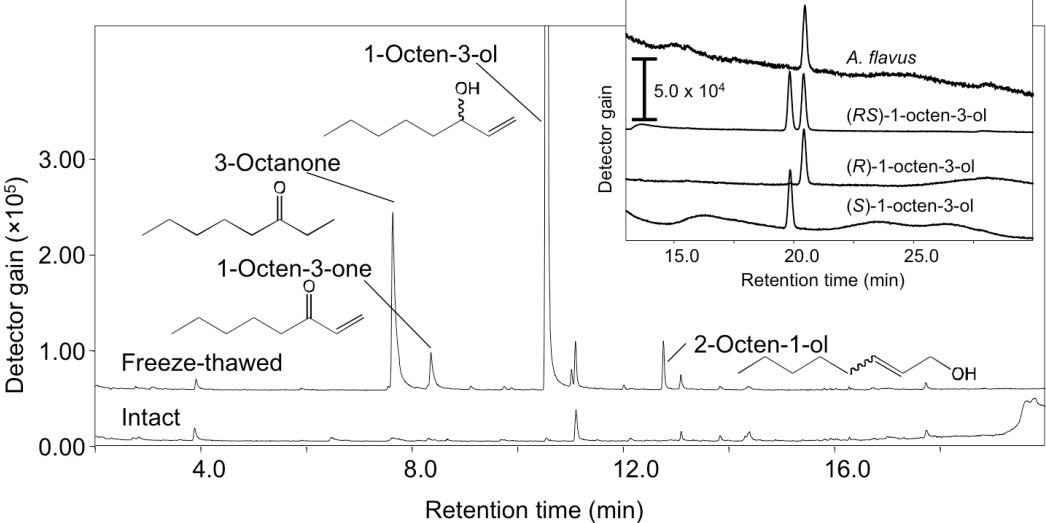

**Figure 1 SPME-GC/MS analysis of volatiles formed from *A. flavus* conidia.** The conidia prepared from 1-week old GMM plates were suspended in GMM liquid medium to be $1.0 \times 10^9$ spores mL$^{-1}$, and immediately (intact) or after one cycle of freeze-thaw treatment (freeze-thawed) the volatiles were collected with SPME fiber at 22 °C, then analyzed with GC-MS. Separation of the enantiomers of 1-octen-3-ol is shown in the inset.

with the amount of 1-octen-3-ol (Fig. 2A). Formation of 1-octen-3-ol was completely suppressed when the freeze-thaw treatment was carried out after heat-treatment (in boiling water for 10 min) on conidia.

## Formation of 1-octen-3-ol from *A. flavus* mutants

In order to estimate an involvement of each *ppo* gene and *lox* gene in 1-octen-3-ol formation, we quantified the volatile formed from conidia of deletion mutants Δ*ppoA*, Δ*ppoC*, Δ*ppoD*, and Δ*lox* (*Brown et al., 2009*). IRT4 that was deficient in expression of all the five oxygenases (Δ*ppoD*; *ppoA*, *–B, and –C*; and *lox IRT*) (*Brown et al., 2009*) was also used. As with the wild type spores, substantial amounts of 1-octen-3-ol were only formed after the freeze-thaw treatment of the mutant conidia. The amount of 1-octen-3-ol in each strain showed no significant difference from that in the wild type strain (Table 1). The profiles of the other C8 volatiles detected with SPME-GC/MS were also similar among all strains.

## Effect of 1-octen-3-ol on germination of conidia

When conidia of *A. flavus* were suspended in GMM liquid medium at $1.0 \times 10^6$ mL$^{-1}$, more than 90% germinated after 9 h at 29 °C (Fig. 3A). The germination rate decreased dependent on spore density, and at $1.0 \times 10^9$ spores mL$^{-1}$ germination was significantly inhibited. We also examined germination inhibition with *A. nidulans* under the same experimental condition, and obtained almost the same results (Fig. S1). With both species, vigorous germination of conidia was restored when conidia at $1.0 \times 10^9$ mL$^{-1}$ incubated for 9 h at 29 °C were diluted to $1.0 \times 10^6$ mL$^{-1}$ with fresh GMM.

Next, we added various amounts of 1-octen-3-ol to the conidial suspension set at $1.0 \times 10^6$ mL$^{-1}$. Germination was not affected when the concentration of 1-octen-3-ol was

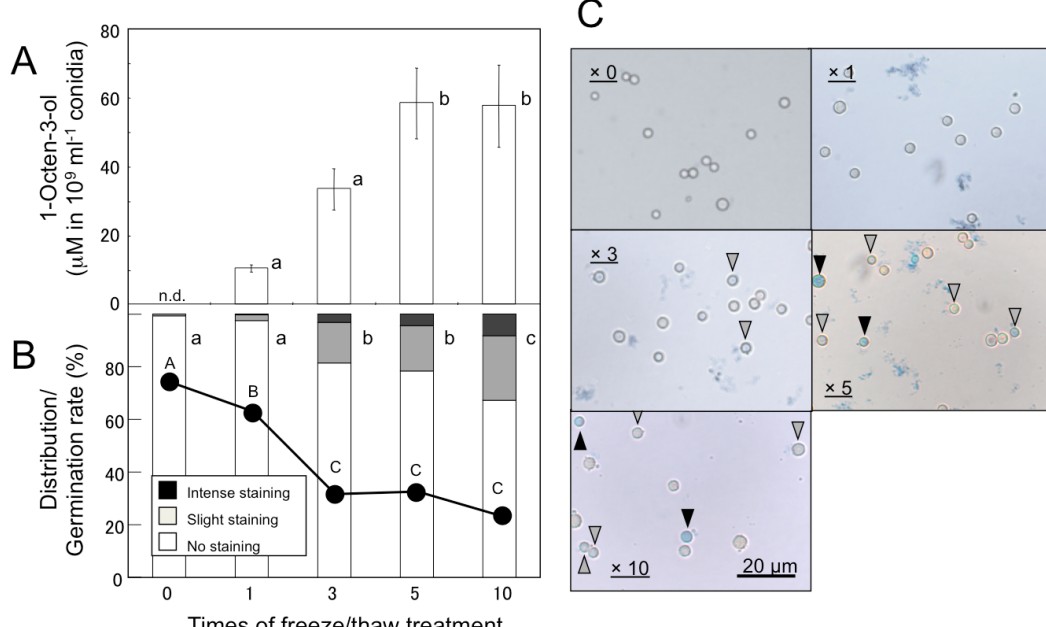

**Figure 2 Effect of freeze/thaw treatment on the amount of 1-octen-3-ol and on the integrity of conidia.** (A) Conidia were collected from *A. flavus* grown for 1-week on GMM plates, and suspended in GMM liquid media to be $1.0 \times 10^9$ spores mL$^{-1}$. They were frozen at $-20\,^\circ$C for 55 min and thawed at 30 $^\circ$C for 5 min. After given cycles of freeze-thaw treatment, 1-octen-3-ol was extracted from the culture with CHCl$_3$/methanol and analyzed with GC/MS. The values are means of three replicates, and error bars show standard error. Different letters indicate statistically significant differences ($P < 0.05$) for the cycles determined by ANOVA (Tukey). (B) The numbers of intensely and slightly stained conidia, and not-stained conidia after Evans Blue staining were counted under microscope, and distribution was calculated. Different letters indicate statistically significant differences ($P < 0.05$, Kruskal-Wallis, $n = 300$). The freeze-thaw-treated conidia were incubated in GMM for 9 h at 29 $^\circ$C, and their germination rates were determined under microscope (shown in B with a line chart). Different letters on the symbols indicate statistically significant differences ($P < 0.05$, Kruskal-Wallis, $n = 500$). (C) Evans Blue staining was performed with conidia served to freeze-thaw treatment for 0, 1, 3, 5, and 10 cycles. Slightly stained and intensely stained conidia are pointed with open and closed triangles, respectively.

**Table 1 The amounts of 1-octen-3-ol formed from the freeze-thaw-treated conidia of *ppo* and *lox* mutants.**

| Strain | 1-Octen-3-ol (nmol mL$^{-1}$)[a] |
|---|---|
| WT | 28.5 ± 6.48 |
| Δ*ppoA* | 39.4 ± 8.74 |
| Δ*ppoC* | 34.1 ± 7.13 |
| Δ*ppoD* | 44.2 ± 13.07 |
| Δ*lox* | 52.1 ± 13.17 |
| IRT4 | 30.7 ± 8.54 |

**Notes.**
[a] The conidia were suspended with GMM to be $1.0 \times 10^9$ ml$^{-1}$, then, the amount of 1-octen-3-ol was determined after one cycle of freeze-thaw treatment. The mean ± SE is shown ($n = 9$). There was no statistically significant difference in the amount of 1-octen-3-ol between each strain (ANOVA, Tukey).

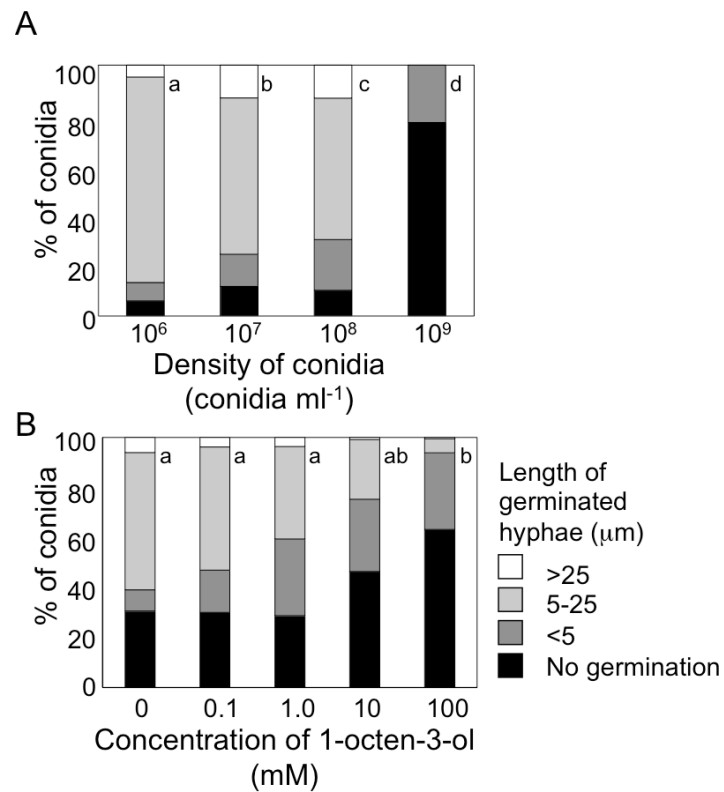

**Figure 3 Germination of conidia and elongation of hyphae were suppressed under high conidial density (A), and in the presence of high concentration of 1-octen-3-ol (B).** (A) Conidia of *A. flavus* were prepared from 1-week old GMM plates, then, resuspended in GMM at $1.0 \times 10^6$ to $1.0 \times 10^9$ spores $mL^{-1}$. The suspensions were incubated at 29 °C for 9 h, then, the germination rate and the length of hyphae of germinated conidia were examined under microscope. (B) To the conidia set at $1.0 \times 10^6$ spores $mL^{-1}$, 1-octen-3-ol was added. Germination of conidia was examined as above. Different letters indicate statistically significant differences ($P < 0.05$, Kruskal-Wallis, $n = 200$).

below 1 mM. At 10 mM germination was slightly inhibited, and a marked inhibition was observed at 100 mM (Fig. 3B). The germination of *A. nidulans* conidia was also inhibited with 1-octen-3-ol at concentration no less than 10 mM (Fig. S1).

### Effect of glucose

We noticed that the concentration of glucose in GMM (initially, 1%) decreased to 0.74 and 0.08% after 9 h-incubation of the conidia at the density of $1 \times 10^8$ $mL^{-1}$ and $1 \times 10^9$ $mL^{-1}$, respectively. Therefore, we anticipated that availability of glucose in the medium was accountable for decreased germination. In order to examine this possibility, the conidia were suspended at $2.5 \times 10^5$ and $2.5 \times 10^8$ $mL^{-1}$ in GMM containing 0 to 4% glucose, and their germination rate was examined (Fig. 4). Without glucose, the conidia showed poor germination at both the densities, but the germination rate at high density was significantly lower than that at low density. Addition of glucose to the media efficiently promoted germination, and at $2.5 \times 10^5$ $mL^{-1}$ addition of 1% glucose efficiently enhanced the germination rate to more than 95%. At higher density of conidia ($2.5 \times 10^8$ $mL^{-1}$), the

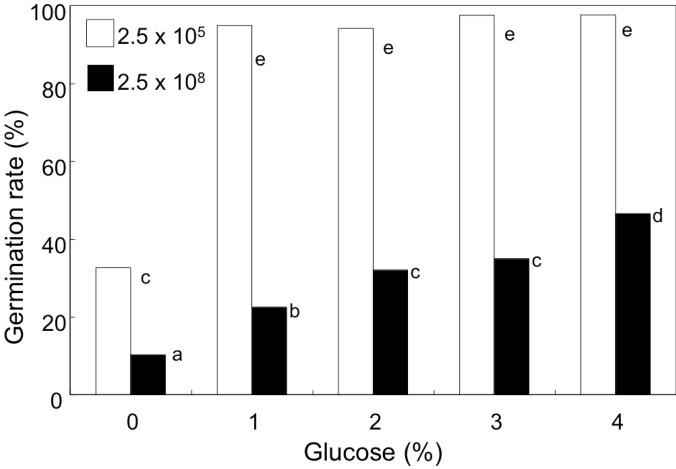

**Figure 4 Effect of glucose concentration on germination of *A. flavus* conidia.** The conidia were suspended with GMM liquid medium containing 0–4% glucose. After incubating at 29 °C for 8.5 h, the germination of conidia was examined under microscope. Different letters indicate statistically significant differences among respective density ($P < 0.05$, Kruskal-Wallis, $n = 300$).

germination rate increased constantly as increasing the initial concentration of glucose, but the rate was still 40% even with 4% glucose.

## Autoinhibitor

When *A. flavus* conidia were suspended with fresh GMM at $1.0 \times 10^6$ mL$^{-1}$, germination of 90% of them was observed after 9 h at 29 °C (Fig. 5). We recovered GMM that had been used to incubate the conidia at $1.0 \times 10^9$ mL$^{-1}$ for 9 h, adjusted its glucose concentration to 1%, and suspended the conidia with the spent medium at $1.0 \times 10^6$ mL$^{-1}$. The germination of conidia was significantly suppressed, and only 40% of them germinated (Fig. 5). When the spent medium was heated with boiling water for 10 min, the germination of conidia recovered, therefore, the factor that suppressed germination was heat-labile.

## DISCUSSION

We found that disintegration of cellular architecture caused by freeze-thaw treatment elicited formation of 1-octen-3-ol in *A. flavus* conidia. When sporophores of mushrooms (*Agaricus bisporus*) were sliced or homogenized, the amount of C8 volatiles increased (*Combet et al., 2009*). *Penicillium roqueforti* also formed significant amount of 1-octen-3-ol after homogenization (*Kermasha et al., 2002*). This suggests that the enzymes are not active in intact tissues/cells of fungi or are lacking accessibility to the substrates (linoleic acid and oxygen), perhaps due to localization of enzyme and substrates in different intracellular compartments. Compartmentalization would be destroyed during loss of cell wall/membrane integrity thus allowing for mixing of enzymes with their substrates or with the factors that might activate the enzymes.

Volatile analysis of oxygenase mutants of *A. flavus* suggested that none of the five oxygenases found in its genome were directly involved in 1-octen-3-ol formation. *A. fumigatus* PpoC forms (*R*)-10-HPO from linoleic acid (*Garscha et al., 2007*) while
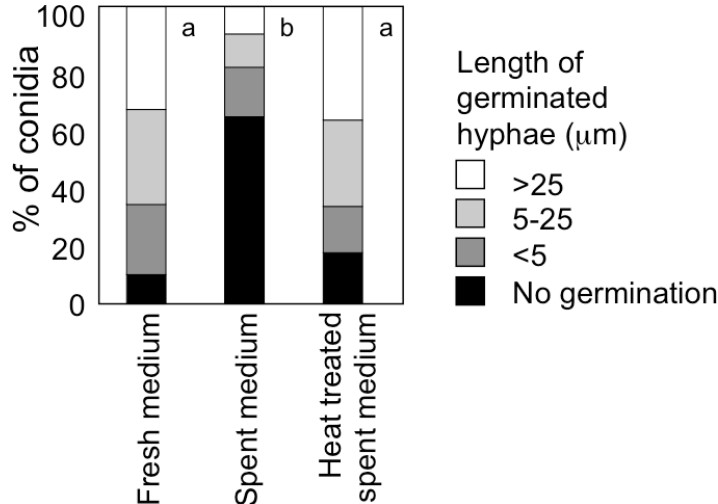

**Figure 5** **Effect of a factor secreted from the conidia at high density.** The medium recovered from the conidial suspension incubated for 9 h at 29 °C at the density of $1.0 \times 10^9$ mL$^{-1}$ (spent medium), the used medium after heat treatment at 100 °C for 10 min (heat treated spent medium), or fresh GMM medium was used to suspend conidia at $1.0 \times 10^6$ mL$^{-1}$. Germination of conidia was examined after incubation at 29 °C for 9 h as above. Different letters indicate statistically significant differences ($P < 0.05$, Kruskal-Wallis, $n = 200$).

($R$)-1-octen-3-ol is formed stereospecifically from ($S$)-10-HPO in mushrooms (*Agaricus bisporus*) (*Wurzenberger & Grosch, 1984*). Even though product specificity of *A. flavus* PpoC has not been identified yet, the discrepancy in stereochemistry further reduces the possibility that PpoC is involved in ($R$)-1-octen-3-ol formation in *A. flavus*. This is unexpected because these are the only genes showing substantial homology to the fatty acid oxygenases involved in oxylipin pathways in animal and plant cells. Because the ability to form 1-octen-3-ol after disruption of conidia was abolished by heat-treatment, there must be another enzyme system(s) that is responsible for 1-octen-3-ol formation. It is known that deletion or overexpression of a *ppo* gene can result in abberant regulation of other *ppo* genes through feedback regulation. For example, overexpression of *ppoA* in *A. nidulans* led to reduced expression of *ppoC*, and deletion of *ppoB* increased expression of *ppoC* (*Brown et al., 2009*; *Tsitsigiannis et al., 2004*). Possibly some unknown feedback mechanism had impact on the volatile production as assessed in this study.

When examining what factors could be involved in germination inhibition at high spore densities, we found that availability of glucose was an important factor as addition of glucose could partially remediate the germination defect. We also found that a heat-labile component in the media recovered from *A. flavus* conidia suspension incubated for 9 h at high density ($1.0 \times 10^9$ mL$^{-1}$) was partly accountable for inhibition of germination. However, in our set up, it did not appear that 1-octen-3-ol was contributing to inhibition of germination. The maximum concentration of 1-octen-3-ol–57.6 µM after 10 cycles of freeze-thaw treatment of $1.0 \times 10^9$ spores/mL$^{-1}$–was well under the 10 mM of exogenous 1-octen-3-ol found to inhibit germination. This is also the case with *A. nidulans* (Fig. S1). From these results, it appears that 1-octen-3-ol is not the principal endogenous

autoinhibitor responsible for inhibiting germination at high spore density for either *A. flavus* or *A. nidulans*. In previous studies showing 1-octen-3-ol suppression of germination of conidia, its amount formed from the conidia had not been determined (*Chitarra et al., 2004*; *Herrero-Garcia et al., 2011*). In order to confirm the role of 1-octen-3-ol in controlling germination of fungal conidia, the amount of 1-octen-3-ol formed by the conidia should be precisely quantified, then, the effect of 1-octen-3-ol at the physiological concentration should be re-evaluated.

In this study, we found that *A. flavus* conidia synthesized C8 volatiles after disruption of their cell integrity. (*R*)-1-Octen-3-ol was most abundantly formed volatile. Analyses on mutant strains revealed that fatty acid oxygenases identified in *A. flavus* so far, namely, *ppoA*, *ppoB*, *ppoC*, *ppoD*, and *lox* were not essential to form 1-octen-3-ol. It is suggested that another, heat-labile enzyme, is involved in 1-octen-3-ol biosynthesis. Even though increasing spore density was correlated with germination inhibition for *A. flavus*, the amount of 1-octen-3-ol formed from the conidia was much lower than that found to inhibit germination of conidia, thus, 1-octen-3-ol is not directly involved in inhibition of germination and the physiological significance of 1-octen-3-ol formation in conidia remains an open question.

### Abbreviations

| | |
|---|---|
| **C8** | carbon-eight |
| **SPME** | solid phase microextraction |
| **Ppo** | Psi-factor producing oxygenase |
| **LOX** | lipoxygenase |

### Funding

This work was supported in part by the Japan Society for the Promotion of Science (JSPS) [KAKENHI (No. 23580151)] and by the Yamaguchi University (Yobimizu Project) to MK, and in part by NSF IOS-0965649 funds to NPK. The funders had no role in study design, data collection and analysis, decision to publish, or preparation of the manuscript.

### Grant Disclosures

The following grant information was disclosed by the authors:
The Japan Society for the Promotion of Science (JSPS): 23580151.
Yamaguchi University (Yobimizu Project).
NSF: IOS-0965649.

### Competing Interests

Nancy P. Keller is an Academic Editor for PeerJ. The other authors declare there are no competing interests.

## Author Contributions

- Kana Miyamoto, Tomoko Murakami and Pattana Kakumyan performed the experiments, analyzed the data, prepared figures and/or tables.
- Nancy P. Keller analyzed the data, contributed reagents/materials/analysis tools, wrote the paper, reviewed drafts of the paper.
- Kenji Matsui conceived and designed the experiments, analyzed the data, wrote the paper, prepared figures and/or tables, reviewed drafts of the paper.

## Supplemental Information

Supplemental information for this article can be found online at http://dx.doi.org/10.7717/peerj.395.

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
