# Peer review of "Formation of 1-octen-3-ol from Aspergillus flavus conidia is accelerated after disruption of cells independently of Ppo oxygenases, and is not a main cause of inhibition of germination"

_PeerJ, doi:10.7717/peerj.395_

## Round 0.1 · original submission · Minor Revisions

Dear Prof. Matsui,

The manuscript has been revised be two expert reviewers in the field . Both reviewers have responded positively although they have made some minor suggestions that once addressed, would result in an improved manuscript.

·

Basic reporting

Generally well presented. No further comments.

Experimental design

The experiments in this paper are well-designed and covered most bases.
Minor comment:
Has the changes of pH in the medium been considered/measured? If there is a change in pH, can the changes in pH affect the germination besides glucose level?

Validity of the findings

The conclusions are generally well-supported.
One minor comment:
On the measuring the level of 1-octen-3-ol in heat-treated conidia, the possibility that the C8 volatiles were evaporated after heat-treatment cannot be ruled out. If this has not been done, the authors should check if there is any loss of 1-octen-3-ol after heat-treatment by adding a small ammount of standard 1-octen-3-ol into the condia before heat-treatment. If there is a loss of C8-volatiles, it will not support the conclusion "there must be another enzyme system(s) that is responsible for 1-octen-3-ol formation".

Additional comments

The hypothesis that C8 volatiles are autoinhibitors for inhibiting germination and the possibility that these C8 volatiles are produced by ppo oxygenases were carefully examined using SPME-GC/MS in combination with well-thought experiments in this paper. The authors showed convincingly that these C8 volatiles are not responsible for the autoinhibition under physiologically relevant context as the level of the C8 volatiles present in the conidia are much lower than the amount required for inhibition of germination. They also showed that the formation of these C8 volatiles are independent of ppo oxygenases by examining the corresponding mutants. The finding in the end about the possibility of an unidentified heat-labile factor may be responsible for the autoinhibition is highly interesting and certainly warrants future investigation.

Minor: It would be helpful if this autoinhibitor can be highlighted in the title as well, although it is acknowledged that the present title is already quite long.

·

Basic reporting

Formation of 1-octen-3-ol from Aspergillus flavus is accelerated after disruption of cells independently of Ppo oxygenases, and is not a main cause of inhibition of germination.
Miyamoto et al.
This contribution describes the crowding effect in conidia of A. flavus (and A. nidulans) in which the presence of high densities of conidia result in inhibition of germination. Addition of glucose to these suspensions partially increased germination. Spent medium of high density spore suspensions also suppressed germination in diluted spore solutions. This suppression activity was absent when the spent medium was boiled. Fresh harvested conidia in high densities were analysed for the presence of the volatile 1-octen-3-ol , but the component could not be detected with GC. After a freeze-thaw treatment small amounts of R-1-octen-3-ol were detected. The amount released by the cells increased after up to 5 freeze-thaw treatments. Three or more freeze-thaw treatments resulted in lowered germination, which was correlated with a slight staining of the cells with Evans Blue. Production of 1-octen-3-ol was similar in mutants that were deleted for fatty acid oxygenases, indicating that 1-octen-3-ol is formed in other ways than via these enzymes. Addition of 10 mM 1-octen-3-ol to conidia showed a marked decrease of germination after 9 h at 29⁰C. The authors conclude that the functioning of 1-octen-3-ol acts as a self-inhibitor of premature germination of conidia remains an open question.

The paper is well written and the sequence of experiments and clear and the work is worthwhile of publication in Peer Journal. The manuscript really adds to the topic. Before, acceptation I would like to see a number of points addressed by the authors.

1. In case of P. paneum, production of 1-octen-3-ol was observed after 22 h in suspensions of 109 conidia/ml in Malt Extract Broth, but no germination-inhibiting activity was observed in fresh suspensions (Chitarra et al. 2004). Do the authors have data if the amount of produced 1-octen-3-ol had increased after a certain period of incubation of suspensions of high conidial density?

2. Germination has dropped with 60% after three or more freeze-thaw treatments, while faint staining with Evans Blue is occurring in less than 20% of the conidia. Is this an indication that the majority of the conidia that do not germinate were not damaged?

3. If used medium of a high density spore solution is boiled, the suppressed germination recovered. If a volatile was present in this solution, could it be evaporated from the solution after heating?

4. Activity of 1-octen-3-ol is tested in solution and millimolar concentrations are needed to inhibit spore germination. The puzzle is if the action as a volatile inhibitor, via the air, is more effective at lower concentrations than in liquid.

Experimental design

See above

Validity of the findings

See above

Additional comments

See above

---

## Round 0.2 · accepted · Accept

All of the minor revisions were adequately addressed and thus manuscript is now acceptable for publication.